# Antibiotic-Resistant *Enterobacteriaceae* in Wastewater of Abattoirs

**DOI:** 10.3390/antibiotics10050568

**Published:** 2021-05-12

**Authors:** Timo Homeier-Bachmann, Stefan E. Heiden, Phillip K. Lübcke, Lisa Bachmann, Jürgen A. Bohnert, Dirk Zimmermann, Katharina Schaufler

**Affiliations:** 1Friedrich-Loeffler-Institut, Institute of Epidemiology, 17493 Greifswald-Insel Riems, Germany; phillipkarl.luebcke@stud.uni-greifswald.de; 2Institute of Pharmacy, University of Greifswald, 17489 Greifswald, Germany; stefan.heiden@uni-greifswald.de (S.E.H.); katharina.schaufler@uni-greifswald.de (K.S.); 3Leibniz-Institut für Nutztierbiologie, Institute of Nutritional Physiology “Oskar Kellner”, 18196 Dummerstorf, Germany; bachmann@fbn-dummerstorf.de; 4Friedrich Loeffler-Institute of Medical Microbiology, University Medicine Greifswald, 17475 Greifswald, Germany; juergen.bohnert@med.uni-greifswald.de; 5Greenpeace e.V., 20547 Hamburg, Germany; dirk.zimmermann@greenpeace.org

**Keywords:** MDR, slaughterhouse, wastewater, resistance, *Enterobacteriaceae*

## Abstract

Antibiotic-resistant *Enterobacteriaceae* are regularly detected in livestock. As pathogens, they cause difficult-to-treat infections and, as commensals, they may serve as a source of resistance genes for other bacteria. Slaughterhouses produce significant amounts of wastewater containing antimicrobial-resistant bacteria (AMRB), which are released into the environment. We analyzed the wastewater from seven slaughterhouses (pig and poultry) for extended-spectrum β-lactamase (ESBL)-carrying and colistin-resistant *Enterobacteriaceae*. AMRB were regularly detected in pig and poultry slaughterhouse wastewaters monitored here. All 25 ESBL-producing bacterial strains (19 *E. coli* and six *K. pneumoniae*) isolated from poultry slaughterhouses were multidrug-resistant. In pig slaughterhouses 64% (12 of 21 *E. coli* [57%] and all four detected *K. pneumoniae* [100%]) were multidrug-resistant. Regarding colistin, resistant *Enterobacteriaceae* were detected in 54% of poultry and 21% of pig water samples. Carbapenem resistance was not detected. Resistant bacteria were found directly during discharge of wastewaters from abattoirs into water bodies highlighting the role of slaughterhouses for environmental surface water contamination.

## 1. Introduction

Antimicrobials are essential to treat bacterial infections. The emergence and spread of antimicrobial resistance (AMR) has been recognized globally as a serious threat to public health [1]. Moreover, antimicrobial consumption can lead to alterations in the human microbiota and select for antimicrobial-resistant bacteria (AMRB). AMRB and partially metabolized antimicrobials are excreted into the wastewater with urine and feces and may end up in surface waters and cropland via the sewage system [2].

Among the most critical antibiotic-resistant bacteria, extended-spectrum β-lactamase (ESBL)-producing Enterobacterales are listed on the WHO’s “Global priority list of antibiotic-resistant bacteria to guide research, discovery and development of new antibiotics” [3]. ESBL-producing bacteria excrete enzymes that hydrolyze 3rd- and 4th-generation cephalosporins, such as cefotaxime, and were described for the first time in 1983 [4]. There are different ESBL variants. The first were TEM and SHV. The prevalence of these enzymes has since declined, while at the same time isolates producing CTX-M-type β-lactamases have spread worldwide. ESBL are plasmid mediated [5]. ESBL-producing Enterobacterales show a high zoonotic potential [3].

Industrial agriculture, in its present form, relies heavily on the widespread use of antimicrobials to improve animal health, welfare and productivity. Antimicrobials are administered to livestock either as therapeutics (to treat individual animals) or in a metaphylactic approach, i.e., the presence of clinical illness in a small number of animals triggers drug administration of the whole herd or flock [6]. 

Although an overall decline in sales of veterinary antibiotics could be observed between 2011 and 2018 [7], the broad administration of antibiotic substances—for example in pig husbandry—is still ongoing. Tetracyclines, amoxicillin, macrolides and colistin are most frequently used in pork production. Studies reported in the German poultry sector indicated that broilers treated for 10 days with one active compound within their 39-day production period in 2011 [8] and between 2014 and 2017 showed no change of antimicrobial usage [9]. 

Of particular concern is the emergence of 3MRGN, i.e., Gram-negative bacteria that are resistant against three of the most important antimicrobial classes. They occur not only in healthcare settings [10], but also in poultry [11,12] and swine production chains [13]. In addition, Enterobacterales often carry mobilizable colistin resistance (*mcr*) genes [14,15]. Colistin resistance is of particular importance, as the agent had to be reintroduced into human medicine—despite nephrotoxic and neurotoxic properties—for therapy of multidrug-resistant *Acinetobacter baumannii*, *Pseudomonas aeruginosa* and carbapenemase-producing *Enterobacteriaceae* [16]. Consequently, the WHO included colistin into the group of the “highest priority critically important antimicrobials” for human medicine [17]. 

Throughout our manuscript, isolates exhibiting resistance to at least three antimicrobial classes are considered to be multidrug-resistant (MDR). 

In 2017, more than 1.5 million tons of poultry and almost 5.5 million tons of pigs were slaughtered in Germany [18]. The slaughtering process produces large amounts of wastewater, potentially contaminated with bacteria resistant against antimicrobials (several 1000 L per 1000 kg live weight) [19]. In this regard, wastewater represents a source for environmental pollution with antibiotic-resistant/MDR bacteria, possibly driven by inadequate treatment in the in-house wastewater treatment plants (WWTPs) [20]. 

The aim of this study was to evaluate the occurrence and diversity of antimicrobial-resistant bacteria (Enterobacterales and *Staphylococcus aureus*) in wastewater of seven slaughterhouses at two time points. Besides species identification, molecular similarity analyses and phenotypic resistance testing were performed. In addition, whole-genome sequencing and bioinformatics analysis were carried out for selected isolates.

## 2. Results

### 2.1. Isolates

For the two sampling time points and all slaughterhouse wastewater outlets, bacterial growth was detected on CHROMID ESBL plates. Growth on the colistin plates was present for all samples obtained in November, but only for two in December (Table 1).

No growth was detected on the culture media specific for MRSA nor for carbapenem-resistant bacteria (NDM/OXA-48) (Table 1). According to color and morphology of the colonies, *Escherichia* (*E*.) coli, *Klebsiella* (*K*.) spp., *Enterobacter* (*E*.) spp. and *Citrobacter* (*C*.) spp. suspect colonies were subcultured, and in total, 55 isolates were further analyzed in the VITEK2 MS system (Table 2).

Putative ESBL-producing *E. coli* isolates were found at both time points from all slaughterhouse wastewater samples. Putative ESBL-producing K. pneumoniae isolates were found in four of seven samples collected in November and only in two of six samples obtained in December (57% and 33%, respectively). On the colistin-containing media, *E. coli* grew in three of seven November samples and only from one slaughterhouse (F) in December. Putative colistin-resistant K. pneumoniae colonies were present only in the samples from slaughterhouse D in November. Moreover, in November, two Raoultella (R.) ornithinolytica isolates were detected in samples from slaughterhouse A. In two samples from November (slaughterhouses E and G) E. cloacae was detected. In December, no samples at all were obtained from slaughterhouse G due to operational reasons (Table 2).

### 2.2. Similarity Analysis 

Based on RAPD, different patterns (“fingerprints”) were detected for the isolates from samples of different slaughterhouse effluents. By contrast, highly similar patterns were regularly found for bacterial isolates from the same slaughterhouse samples at the individual sampling times. Highly similar patterns were shared between the duplicate samples as well as between samples originating from different selective plates (ESBL and COLISTIN). This was most pronounced for *E. coli* from slaughterhouse F, with four of five *E. coli* showing highly similar patterns (Figure 1). Note, however, that no consistent patterns were evident for bacterial isolates between the two sampling points.

### 2.3. Antimicrobial Susceptibility Testing (AST)

Except for the two *R. ornithinolytica* strains, all isolates were resistant to piperacillin. Furthermore, the majority of isolates were additionally resistant to the 2nd- and 3rd-Gen cephalosporins cefuroxime (95% [n = 52]), cefotaxime (89% [n = 49]) and ceftazidime (95% [n = 52]). Twenty-nine isolates were resistant to ciprofloxacin (53%), while five isolates (9%) were resistant to gentamicin and 26 (47%) were resistant to tetracycline. All isolates were susceptible to meropenem and imipenem. Twenty (36%) isolates exhibited a resistance against colistin, which is a last-resort antibiotic. It is noticeable that 43 (78%) of 55 isolates examined exhibited resistance against three or more antimicrobial classes and can, therefore, be specified as phenotypically multidrug-resistant (MDR). For all *K. pneumoniae* (n = 10) and 31 *E. coli* (78%) isolates, a MDR phenotype was determined, respectively. Twenty-three MDR *E. coli* were obtained in November 2020, while eight were isolated in December 2020. Similarly, MDR *K. pneumoniae* were mainly obtained in November (seven isolates vs. three isolates in December). Details are given in Table 3.

### 2.4. Whole-Genome Sequencing

Two different phylotypes were identified among the three whole-genome sequenced *E. coli* isolates. The isolates with the numbers 469 and 472 belong to phylotype A and the isolate 622 to phylotype B1. The following sequence types (ST) appeared: *E. coli*: ST4981 (isolates 469, 472) and ST533 (isolate 622); K. pneumoniae: ST29 (isolate 558); C. freundii: ST248 (isolate 620). Genes encoding for various proteins associated with antibiotic resistance, e.g., bla_CTX-M-15_, tet(34) and mcr-1.1, were detected, thus matching our phenotypic findings. To further compare phenotypic and genotypic resistance data, the antibiotic classes aminoglycosides, β-lactams and polypeptides (colistin) were included in our subsequent analysis (Figure 2). For colistin resistance, our pheno- and genotypic results concorded. Possessors of mcr-1.1 (numbers: 469; 472; 622) were phenotypically colistin-resistant, whereas bacteria without the resistance gene (numbers: 558; 620) were susceptible. Similarly, different genes encoding β-lactam resistance, such as bla_CTX-M-15_, bla_TEM-181_ and bla_SHV-187_ occurred. Phenotypically, all five isolates were resistant against β-lactam antibiotics. In addition, various resistance genes encoding for multiple drug resistance were found (e.g., marA, tolC, emrK). 

To gain insight into the virulence of the selected isolate, we investigated virulence-associated genes related to pathogenic bacteria. The main focus laid on hypermucoviscosity, iron acquisition and resistance against heavy metals and biocides. In relation to hypermucoviscosity, we found rmp, which is associated with hypermucoviscosity and different genes which play a role in iron re-uptake, such as yersiniabactin, enterobactin, and aerobactin synthesis (irp1, irp2, fyuA, ybtAEPQSTUX (469, 472, 558), entABCDEFS (all isolates; but 558 and 620 missing entD), and iucABCD (622), iutA (558, 620, 622)). Furthermore, we noticed genes conferring resistance against heavy metals (zntAR, arsB (all isolates), czc system (469, 472, 622), pcoABCDERS (620), silABCEFPRS (620)) and biocides (emrE (620, 622), mdfA (469, 472, 620, 622), ydgEF(mdtJI) (469, 472, 622), qacF (469, 472), baeSR (all isolates)) transpired. We detected several extra-intestinal pathogenic E. coli (ExPEC)-typical virulence-associated genes e.g., iutA (558, 620,622), fimC (558, 622), fyuA (469, 472, 558) and irp2 (469, 472, 522). Some of them were present in all three E. coli isolates (malX, ibeB/C, iss2) (details are given in Figure 2).

## 3. Discussion

We investigated wastewater from seven German slaughterhouses at two time points for the occurrence of antibiotic-resistant bacteria. Our results clearly demonstrate that these germs are present in wastewater, and thus may lead to contamination of downstream water bodies and agricultural settings. Since antibiotic-resistant bacteria apparently survive the passage through the in-slaughterhouse treatment plant, the procedure may not be suitable to prevent bacterial spill-over effects into the environment. 

The presence of enterobacteria in wastewater is not surprising as most of them are part of the healthy animal microbiota and are able to colonize the gastrointestinal tract. Savin et al. recently demonstrated the presence of such bacteria in two German poultry and pig slaughterhouses. The authors examined water samples along the process chain for ESBL-producing Enterobacterales and colistin-resistant bacteria in two poultry and two pork slaughterhouses. They found the respective pathogens on all levels [13,14,20,21]. Note, however, that in this study, only a small proportion of the investigated isolates originated from wastewater. Here, we received a total of 55 ESBL-carrying or colistin-resistant isolates. 

Two sources for the isolated pathogens can be considered. On the one hand, a slaughterhouse-specific microbial flora could have established itself, from which bacteria are continuously washed off and released into the environment. On the other hand, the microorganisms could originate from the animals currently slaughtered. To test this, we compared the isolates from the two time points of the respective slaughterhouses. The fact that RAPD analysis regularly revealed highly similar patterns within a sampling location at one sampling time point, but never disclosed matches in patterns between sampling events at one location may suggest that the slaughtered animal species is crucial to the bacterial composition of the wastewater released by the abattoir.

### 3.1. Resistance Data

Overall, we detected high levels of MDR bacteria in all slaughterhouse wastewater samples. All *K. pneumoniae* isolates were multidrug-resistant and 78% of *E. coli* samples demonstrated such phenotypes. In wastewater samples from poultry slaughterhouses, we found 100% of 25 examined isolates to be MDR (19 *E. coli* and six *K. pneumoniae*), i.e., isolates exhibited resistance against three or more antimicrobial classes. Thus, our results match recently published findings, reporting a similarly high level of MDR bacteria overall [13,14,20,21]. 

Of 25 *E. coli* and *K. pneumoniae* isolates in wastewater samples from four pig slaughterhouses, 64% were MDR. For *E. coli* 57% (12 of 21) isolates exhibited a MDR phenotype, and thus, we can confirm the results of Savin et al. [21]. To the best of our knowledge, we provide here data on the occurrence of MDR *K. pneumoniae* in wastewater from German pig slaughterhouses for the first time. All *K. pneumoniae* isolates investigated were MDR (n = 4). The significantly lower proportion of MDR bacteria in wastewater from pig slaughterhouses in our study may be due to the lower frequency of antibiotic treatment in pigs [7,8,9,22]. 

All isolates were susceptible to the carbapenems meropenem and imipenem. Studies from Asia and Africa report high levels of carbapenem-resistant Enterobacterales [23,24]. These findings are not surprising, since carbapenems are not approved for use in livestock in the European Union (EU) and similar results have been reported by others (reviewed in [25]). In contrast, 20 of the 55 examined isolates (36%) exhibited resistance against colistin, which is a last-resort antibiotic for humans. Colistin has been used in veterinary medicine in recent decades in the EU, leading to selective pressure and the evolution of colistin-resistant bacteria [26]. Colistin is predominantly used in pig and cattle production to control enteric infections caused by *E. coli* and *Salmonella* or for metaphylactic treatment [27,28], while for the poultry sector, there are no relevant indications other than colibacillosis [29]. Similar to the prevalence of MDR bacteria, differences in the proportions of colistin-resistant *Enterobacteriaceae* between samples from poultry and swine slaughterhouses were observed. For 25 pig slaughterhouse isolates (21 *E. coli* and four *K. pneumoniae*) 16% (n = 4) were colistin-resistant (one *E. coli* and three *K. pneumoniae*), while of 25 analyzed poultry slaughterhouse isolates, 56% (n = 14) were resistant against this last-resort antibiotic. Both values are significantly higher than those previously reported for the pig and poultry sectors, respectively. Irrgang et al. published colistin resistance data for *E. coli* isolates between 2010 and 2015 of about 5% in broilers, 10% in turkeys and approx. 3% in pigs [30]. However, in a recently published study by Savin et al., similar orders of magnitude of colistin-resistant *E. coli* along the slaughterhouse process chain (pig: 45.0%, poultry: 52.8%) were reported. The authors mention that this might be due to co-selection through other antimicrobials that are often used in pigs, e.g., macrolides, lincosamides and tetracyclines or a linkage to their selective isolation procedure [21]. 

### 3.2. Sequencing Data

Analysis of the whole-genome sequence data revealed broad agreement of phenotypic and genotypic results. For the phenotypic resistances observed, genotypic resistance determinants were always found as well (Figure 2). Details of the individual whole-genome sequenced *E. coli* are summarized below.


*E. coli ST4981*


In a recent investigation of irrigation water in Switzerland with a focus on antibiotic-resistant bacteria, ESBL-producing *E. coli* of sequence type ST4981 were detected, among others [31]. For this ST, only the virulence genes *gad* (glutamate decarboxylase) and *iss* (increased serum survival) were detected. These two virulence factors were also present in our two whole-genome sequenced ST4981 isolates in addition to others (Figure 2). Additionally, *bla*_CTX-M-15_ was detected in all isolates. CTX-M-15 has been reported globally in all major ecological niches (humans, animals, and environment). CTX-M-15 is an excellent example for the public health threat that involves the circulation of resistant Enterobacterales in different ecological niches in the “One Health” context [32]. The combination of ST4981 and CTX-M-15-carrying *E. coli* has already been detected in fecal samples from piglets with diarrhea [33]. Both of our isolates belonged to phylogroup A, which is predominantly associated with commensal *E. coli* [34].


*E. coli ST533*


In 2009, Bonnedahl et al. detected ST533 in a sample of “offshore” gulls [35]. The authors conclude that higher resistance levels in “city-dump” gulls can be explained by greater exposure to human activities, especially higher antibiotic pressure. In a study of ESBL/AmpC-producing *E. coli* obtained along the poultry chain between 2008 und 2013, two isolates of sequence type ST533 were found. 

ST533 was also detected as part of a study of urine samples from dogs and cats in Switzerland. Like the ST533 isolate of the present study, the corresponding isolate was assigned to phylogroup B1 (commonly associated with commensal strains [34]) and carried a *bla* gene of the CTX-M type, too. However, unlike the present isolate, it was CTX-M-15 [36]. *bla*_CTX-M-27_ is a CTX-M type that has recently been found increasingly. It is present in humans, animals, and environment, especially in *E. coli* ST131 [37,38,39,40] and also in the ST533 isolate of the present study. Interestingly, occurrence of CTX-M-27 has been reported in a study on ESBL-producers in slaughterhouse workers. Dohmen et al. found CTX-M-27 in combination with plasmid rep/inc-types IncFIA-FIB-FII [41], thus matching our findings (IncFIA-FIB-FII).


*K. pneumoniae ST29*


Sequence type 29 has already been identified in *K. pneumoniae* in several studies in different regions of the world in samples of animal origin. These include slaughterhouse effluents in Pakistan [23] and poultry meat imported from the Netherlands in Ghana [24]. In the Pakistan study, nearly 10% of *K. pneumoniae* isolates were carbapenem-resistant, and ST29 accounted for over 50%. However, all isolates were colistin-sensitive. In the Ghanaian study, all *K. pneumoniae* isolates were carbapenem-sensitive regardless of their origin (predominantly The Netherlands, Brazil, and the U.S.). The ST29 isolate of the present study, like the ST29 representatives of the above studies, carried a *bla* gene of CTX-M-15 type and was carbapenem-sensitive, like the Ghanaian (Dutch) isolate. However, it showed phenotypical and genotypical colistin resistance, which was not addressed in the Ghanaian study. 

Hypervirulence in *K. pneumoniae* (hvKP) has been reported worldwide as a contributor to serious infections and outcomes in pathogens, both in hospitals and the community [42] and is often accompanied with a hypermucoviscous phenotype that confers resistance to phagocytosis and intracellular killing [43]. Moura et al. reported a ST29 isolate exhibiting a hypermucoviscous phenotype. In addition, in that strain, the same plasmids replicons as in the ST29 isolate of the present study were detected (IncFII and IncFIB) [44]. Therefore, we decided to perform a hypermucoviscosity test with all *K. pneumoniae* isolates. However, none of our isolates exhibited a hypermucoviscous phenotype. 


*C. freundii ST248*


The *C. freundii* isolate in the present study had only a few resistance determinants (*bla*_CMY-77_, *qnrB18*). So far, no reports are available linking this ST to clinical manifestations or remarkable resistance traits. Only in a study from China—dealing with carbapenem-resistant *Enterobacteriaceae*—was this ST detected. However, the *C. freundii* ST248 isolates in that study were carbapenem-susceptible [45]. 

### 3.3. Wastewater Treatment

According to the Directive 2010/75/EU of the European Union on industrial emissions (integrated pollution prevention and control), the use of best available techniques is prescribed for the treatment of wastewater and according to Decision (EU) 2016/902 of the European Union [46], this means that the following requirements are imposed on the wastewater for the point of discharge into the water body: Biochemical Oxygen Demand in 5 days (BOD5), Chemical Oxygen Demand (COD), Ammonium Nitrogen (NH4-N), Total Nitrogen and Total Phosphorus. Microbiological requirements are not specified. 

Recently, Pärnänen et al. showed that wastewater treatment decreased antimicrobial resistance genes (ARG) [47]. They also found lower levels of ARG in effluents from northern European countries compared to southern European countries. Reasons for this may be differences in antibiotic consumption and temperature. However, this study aimed at the detection of ARG in wastewater samples and not on the examination of individual isolates. 

In contrast, other “isolate-based” studies showed that WWTP do not lead to the reduction of antimicrobial-resistant bacteria in wastewater or achieve this task only insufficiently [2,48]. The present study lacks insight into the condition prior to wastewater treatment but the numerous detections of multidrug-resistant pathogens of various species after treatment suggest that the reduction of antibiotic-resistant bacteria may be insufficient. 

Karkman et al. reviewed that urban WWTPs are among the main sources of both AMRB and antibiotic-resistance genes since they provide unique interfaces between human society and the environment as sewage from households and hospitals contain antibiotics and bacteria of human origin, potentially providing a selective pressure for AMRB and ARGs prior to their release into the environment [49]. While hardly any studies have been published on slaughterhouse WWTP, it must be assumed that similarly favorable conditions for the development of resistant bacteria exist in wastewater released from abattoirs. Some authors report that the conditions in urban WWTPs may be favorable for the selection of ARB which, in turn, can transfer the resistance determinants to susceptible bacteria (reviewed in [49,50]), indicating the need for further research on this topic. These concerns mentioned should be addressed before large-scale investments are made in wastewater treatment plants worldwide. 

## 4. Materials and Methods

### 4.1. Sampling Locations and Sample Collection

Samples were collected by two Greenpeace e.V. collection teams from Greenpeace e.V. on the same two days in November and December 2020, respectively. They were taken from seven slaughterhouses (four pig and three poultry slaughterhouses) directly from the outlet of the WWTP of the slaughterhouse, i.e., immediately before discharge to the receiving water body/river (for slaughterhouse E the outlet of the effluent was below the water surface in the receiving water body). Approximately 500 mL were collected in a sterile plastic container (Carl Roth GmbH, Karlsruhe, Germany) and transported refrigerated to the Friedrich-Loeffler-Institute (FLI). Each location was sampled at two different times (two days) and samples were taken in duplicate each. In December, no samples could be obtained from Slaughterhouse G due to operational reasons (26 samples total).

### 4.2. Isolation of Bacteria and Identification 

In the laboratory, 100 mL of water samples were first pre-filtered using a sterilized gauze (PZN: 04046708, FESMED Verbandmittel GmbH, Frankenberg/Sa., Germany) to remove the fine particles. The pre-filtered water was free from any macro-particles, sediment and most fine particles. Thereafter, each sample was filtered using the EZ-Fit filtration system with 0.45 µm pore size filter membranes (merckmillipore, Darmstadt, Germany). After filtration, filter membranes were transferred to 10 mL TSB (Carl Roth GmbH, Karlsruhe, Germany) containing 2 µg/mL cefotaxime (VWR International, Darmstadt, Germany) followed by overnight incubation at 37 °C and shaking at 200 rpm. Depending on the turbidity level, dilution of the overnight cultures followed (up to 10,000 fold dilution). For each sample, 100 µL of the dilutions were plated on chromogenic media CHROMID CARB/OXA, CHROMID ESBL, CHROMID Colistin, and CHROMID MRSA agar plates (bioMérieux, Nürtingen, Germany), and incubated overnight at 37 °C. Putative antibiotic-resistant colonies of *E. coli* (red-purple colonies) and KEC (*Klebsiella* spp., *Enterobacter* spp., *Citrobacter* spp. (blue colonies)) were subcultivated until pure cultures were achieved. Single pure colonies were picked for further verification and characterization.

### 4.3. Antimicrobial Susceptibility Testing (AST)

Bacterial species were initially identified by MALDI-TOF MS (VITEK2 MS, bioMérieux, Nürtingen, Germany). AST was carried out using VITEK2 (bioMérieux, Nürtingen, Germany). Testing was performed using software version 9.02 and AST-N223 card, according to the manufacturer’s instructions. The AST card used for the VITEK2 included an ESBL confirmation test. Second- and third-generation cephalosporins (ceftazidime, cefotaxime and cefuroxime) were used alone or in combination with clavulanic acid. A reduction of growth in the presence of clavulanic acid was considered indicative of ESBL production.

In addition, 96-well plate broth microdilution was performed for determination of minimal inhibitory concentrations (MICs) of veterinary important antimicrobial compounds, e.g., colistin. A single inoculum adjusted to a McFarland standard of 0.5 in 0.9% NaCl was used. Fifty µL of the suspension were diluted in 11.5 mL of Mueller-Hinton II broth (Oxoid, Wesel, Germany). One hundred microliters of the broth were then transferred to Merlin MICRONAUT S 96-well antimicrobial susceptibility testing plates for farm animals (“Großtiere 4”) (Merlin, Bornheim-Hersel, Germany). The results were assessed visually after incubation at 37 °C for 18–24 h. Growth was considered when turbidity was present at the bottom of the well. The tests were considered as valid only if growth in the internal growth control was observed.

MIC Breakpoints were set according to the European Committee on Antimicrobial Susceptibility Testing (EUCAST) breakpoint tables for interpretation of MICs and zone diameters (Version 11.0, 2021. http://www.eucast.org) (last accessed on 8 May 2021).

### 4.4. Similarity Analysis

The randomly amplified polymorphic deoxyribonucleic acid analysis by PCR (RAPD-PCR) involves the amplification of random segments of genomic DNA using short arbitrary primers. It is used for bacterial identification on the strain or isolate level [51,52]. DNA was extracted using the MasterPure™ DNA Purification Kit for Blood, Version II (Lucigen, Middleton, WI, USA). PCR reactions included 2.5 μL of DNA template mixed with 10 μL primer 1 (5′-AAGAGCCCGT-3′) and 2 (5′-GCGATCCCCA-3′) (10 μ/L) (Eurofins, Hamburg, Germany), 12.5 μL nuclease-free water (ad 50 μL) and 25 μL Master Mix (DreamTaq™ Green PCR Master-Mix, Fisher Scientific, Schwerte, Germany). PCR was performed under the following cycling conditions: initial denaturation 15 min @ 95 °C; 35 cycles of denaturation 1 min @ 94 °C, annealing 1 min @ 36 °C and elongation 2 min @ 72 °C and a final elongation 10 min @ 72 °C. Amplified products were separated by electrophoresis on 1.5% agarose gels and stained with GelRed (VWR International, Darmstadt, Germany). A 1 kb ladder (VWR International, Darmstadt, Germany) was used as molecular mass marker. Patterns were determined visually.

### 4.5. Hypermucoviscosity

Hypermucoviscosity experiments were performed using the string test. Strings of five mm or longer, which formed after stretching on the tip of a sterile inoculation loop were defined positive [42]. Experiments were performed with three technical replicates and three biological replicates.

### 4.6. Sequence Analysis

According to the resistance profiles, five particularly resistant isolates were selected and subjected to whole-genome sequencing (WGS). Sequences of three *E. coli*, one *C. freundii*, and one *K. pneumoniae* were generated using the Illumina NextSeq 550 platform (Microbial Genome Sequencing Center, Pittsburgh, PA, USA [MiGS]). DNA was extracted using the MasterPure™ DNA Purification Kit for Blood, Version II (s. above). After quantification and initial quality control, DNA was shipped to MiGS. MiGS performed library preparation as published elsewhere [53] followed by sequencing using 2 × 150 bp paired-end reads. 

Raw sequencing reads were adapter-trimmed (k-mer-based trimming using 23-mers down to 11-mers at the right end using the included adapter references; additional trimming by paired read overlap), contaminant-filtered (k-mer-based removal of phiX174 sequences), and quality-trimmed (trimming on both sites for regions with quality < 3; removal of poly G tails ≥ 10 bp; maximum number of Ns after trimming: 0; minimum average quality after trimming: 18; minimum length: 32 bp, filtering reads with entropy below 0.5 to remove low-complexity reads) using BBDuk from BBTools v. 38.89 (http://sourceforge.net/projects/bbmap/) (accessed on 1 February 2021). Both trimmed reads and raw reads were quality controlled using FastQC v. 0.11.9 (http://www.bioinformatics.babraham.ac.uk/projects/fastqc/) (accessed on 1 February 2021). De novo genome assemblies were conducted by employing the assembly pipeline shovill v. 1.1.0 (https://github.com/tseemann/shovill) (accessed on 1 February 2021) in combination with SPAdes v. 3.15.0 [54]. As part of the pipeline, trimmed reads were subsampled to assemble at a maximum coverage of 100×. Besides the polishing step as part of the shovill pipeline, assemblies underwent an additional polishing step. For this, all trimmed reads were mapped back to the contigs using BWA v. 0.7.17 [55]. The obtained SAM/BAM files were sorted and duplicates marked with SAMtools v. 1.11 [56]. Finally, variants were called with Pilon v. 1.23 [57]. The polished assemblies were checked for suspicious assembly metrics (e.g., high count of contigs and genome size, high N50/N90, low L50/L90). Additionally, CheckM v. 1.1.3 [58] was employed to estimate genome completeness and contamination. The in-silico multi-locus sequence typing (MLST) and antibiotic resistance/virulence gene detection were carried out using mlst v. 2.19.0 (https://github.com/tseemann/mlst) (accessed on 1 February 2021) and ABRicate v. 1.0.0 (https://github.com/tseemann/abricate) (accessed on 1 February 2021), respectively. Both tools rely on 3rd party public databases (e.g., PubMLST [59], VFDB [60], ResFinder [61], PlasmidFinder [62], BacMet [63], ARG-ANNOT [64], Ecoli_VF (https://github.com/phac-nml/ecoli_vf) (accessed on 1 February 2021)).

## 5. Conclusions

Even if a reduction during wastewater treatment occurs, it has to be assumed that significant bacterial loads are present in the production areas of the slaughterhouses. This fact is alarming in two respects: first, it results in the possibility of the pathogens entering downstream water sources and, thus, the environment and the food chain and second, the pathogens pose a risk to the employees of the slaughterhouse.

## Figures and Tables

**Figure 1 antibiotics-10-00568-f001:**
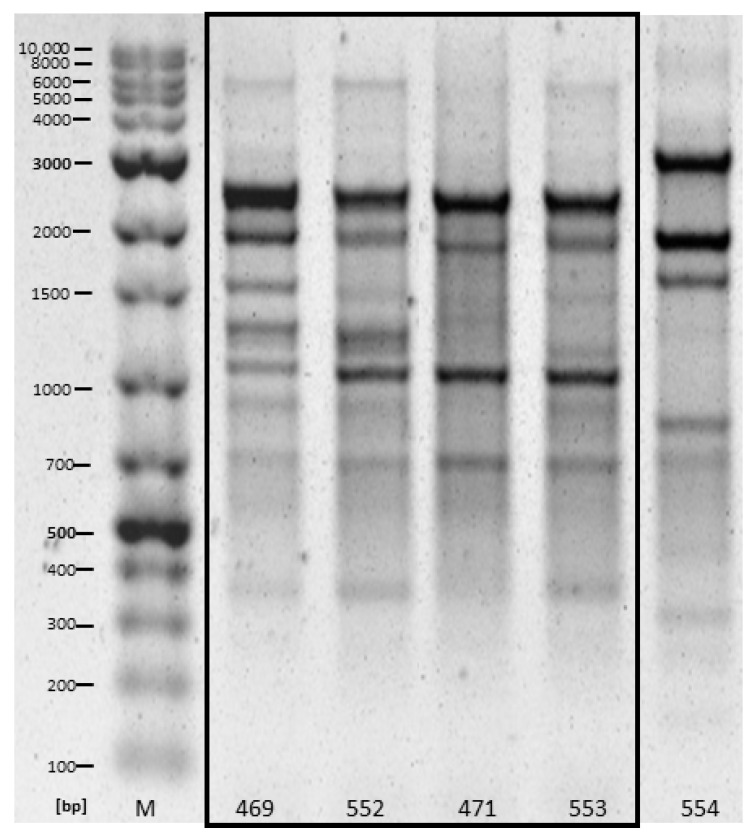
RAPD patterns of five *E. coli* isolates obtained from slaughterhouse D in November 2020, highly similar patterns are framed by the box. Samples 469 and 471 originated from COLISTIN plates (duplicate 1 and 2, respectively), while 552 and 553 originated from ESBL plates (duplicate 1 and 2, respectively). Sample 554 originated from an ESBL plate of duplicate 2.

**Figure 2 antibiotics-10-00568-f002:**
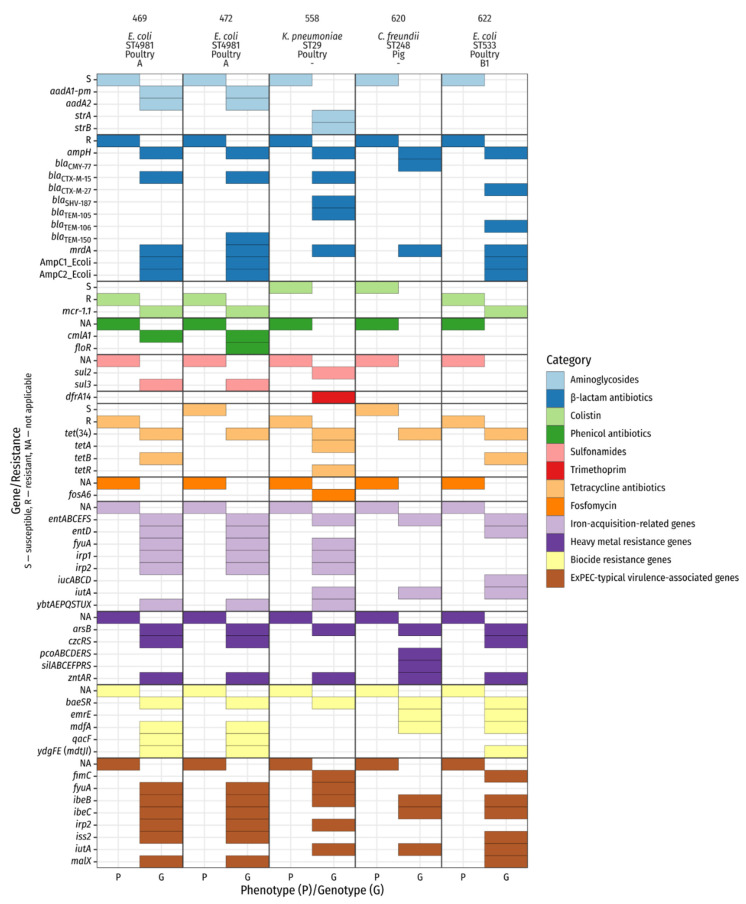
Phenotypic and genotypic characterization of sequenced isolates. Genotypic data represent selected results from ABRicate (ARG-ANNOT, BacMet, Ecoli_VF). The header lists isolate number, species, sequence type, origin of isolation, and *E. coli* phylotype. *mrdA* is annotated as Penicillin_Binding_Protein_Ecoli in the ARG-ANNOT database.

**Table 1 antibiotics-10-00568-t001:** Growth on selective media.

Slaughterhouse	Sampling Date	Selective Media				
		ESBL	COL	MRSA	Carb	OXA-48
A (swine)	November 2020	+	+	-	-	-
	December 2020	+	-	-	-	-
B (swine)	November 2020	+	+	-	-	-
	December 2020	+	-	-	-	-
C (swine)	November 2020	+	+	-	-	-
	December 2020	+	-	-	-	-
D (poultry)	November 2020	+	+	-	-	-
	December 2020	+	+	-	-	-
E (swine)	November 2020	+	+	-	-	-
	December 2020	+	+	-	-	-
F (poultry)	November 2020	+	+	-	-	-
	December 2020	+	-	-	-	-
G (poultry)	November 2020	+	+	-	-	-
	December 2020	N/A	N/A	N/A	N/A	N/A

**Table 2 antibiotics-10-00568-t002:** Bacteria with antimicrobial resistance confirmed by Vitek 2 MS and selective media of origin.

Slaughterhouse	Sampling Date	Selective Media	
		ESBL	COL
A (swine)	November 2020	*E. coli* (2), *K. pneumoniae* (2)	*R. ornithinolytica* (2)
	December 2020	*E. coli* (2)	
B (swine)	November 2020	*E. coli* (5)	
	December 2020	*E. coli* (1)	
C (swine)	November 2020	*E. coli* (2)	
	December 2020	*E. coli* (2)	
D (poultry)	November 2020	*E. coli* (4), *K. pneumoniae* (1)	*E. coli* (2), *K. pneumoniae* (1)
	December 2020	*E. coli* (1), *K. pneumoniae* (1)	
E (swine)	November 2020	*E. coli* (3), *E. cloacae* (1)	
	December 2020 *	*E. coli* (4), *K. pneumoniae* (2)	
F (poultry)	November 2020	*E. coli* (2), *K. pneumoniae* (2)	*E. coli* (2)
	December 2020	*E. coli* (2)	*E. coli* (2)
G (poultry)	November 2020	*E. coli* (2), *K. pneumoniae* (1), *E. cloacae* (1)	*E. coli* (2)
	December 2020	N/A	N/A

* In December 2020 *C. freundii* was isolated from colistin-containing medium.

**Table 3 antibiotics-10-00568-t003:** Results of the VITEK2 MS (species allocation), phenotypic resistance profiles of 55 putative ESBL-producing isolates and derived confirmation of an ESBL production as well as MDR status. Sampling months were November (N) and December (D) 2020. Selective media of origin were ESBL (E) and COLISTIN (C). Isolates subjected to whole genome sequencing are indicated with asterisks.

Designation	Slaughterhouse	Sampling Month	Medium	Duplicate	Species	ESBL Production	MDR	Piperacillin	Cefuroxime	Cefotaxime	Ceftazidime	Meropenem	Imipenem	Gentamicin	Ciprofloxacin	Colistin	Tetracycline
467	A	N	C	1	*R. ornithinolytica*	**-**	**-**	S	S	S	S	S	S	S	S	R	S
468	A	N	C	2	*R. ornithinolytica*	**-**	**-**	S	S	S	S	S	S	S	S	R	S
469 *	D	N	C	1	*E. coli*	**+**	**+**	R	R	R	R	S	S	S	R	R	R
470	D	N	C	2	*K. pneumoniae*	**+**	**+**	R	R	R	R	S	S	S	R	R	S
471	D	N	C	2	*E. coli*	**+**	**+**	R	R	R	R	S	S	S	R	R	R
472 *	F	N	C	1	*E. coli*	**+**	**+**	R	R	R	R	S	S	S	R	R	S
473	F	N	C	2	*E. coli*	**-**	**+**	R	S	S	S	S	S	S	R	R	S
474	G	N	C	1	*E. coli*	**+**	**+**	R	R	S	R	S	S	S	R	R	S
475	G	N	C	2	*E. coli*	**+**	**+**	R	R	S	R	S	S	S	S	R	S
545	A	N	E	1	*E. coli*	**+**	**-**	R	R	R	R	S	S	S	S	S	S
546	A	N	E	2	*E. coli*	**+**	**-**	R	R	R	R	S	S	S	S	S	S
547	B	N	E	1	*E. coli*	**+**	**+**	R	R	R	R	S	S	S	S	S	R
548	B	N	E	2	*E. coli*	**+**	**-**	R	R	R	R	S	S	S	S	S	S
549	B	N	E	2	*E. coli*	**+**	**+**	R	R	R	R	S	S	S	R	S	R
550	C	N	E	1	*E. coli*	**+**	**+**	R	R	R	R	S	S	S	R	R	R
551	C	N	E	2	*E. coli*	**+**	**+**	R	R	R	R	S	S	R	S	S	S
552	D	N	E	1	*E. coli*	**+**	**+**	R	R	R	R	S	S	S	R	R	R
553	D	N	E	2	*E. coli*	**+**	**+**	R	R	R	R	S	S	S	R	R	R
554	D	N	E	2	*E. coli*	**+**	**+**	R	R	R	R	S	S	S	R	R	S
555	E	N	E	1	*E. cloacae complex*	**+**	**+**	R	R	R	R	S	S	R	R	S	S
556	E	N	E	2	*E. coli*	**+**	**+**	R	R	R	R	S	S	S	R	S	R
557	F	N	E	1	*E. coli*	**+**	**+**	R	R	R	R	S	S	S	R	S	S
558 *	F	N	E	1	*K. pneumoniae*	**+**	**+**	R	R	R	R	S	S	S	R	S	R
559	F	N	E	2	*E. coli*	**+**	**+**	R	R	R	R	S	S	S	R	S	R
560	F	N	E	2	*K. pneumoniae*	**+**	**+**	R	R	R	R	S	S	S	R	S	R
561	G	N	E	1	*E. coli*	**+**	**+**	R	R	S	R	S	S	S	S	R	S
562	G	N	E	1	*K. pneumoniae*	**+**	**+**	R	R	R	R	S	S	S	R	S	S
563	G	N	E	2	*E. cloacae complex*	**+**	**+**	R	R	R	R	S	S	S	R	S	R
564	G	N	E	2	*E. coli*	**+**	**+**	R	R	R	R	S	S	S	R	S	S
595	E	D	E	2	*E. coli*	**+**	**+**	R	R	R	R	S	S	R	S	S	S
618	F	D	C	1	*E. coli*	**+**	**+**	R	R	R	R	S	S	S	S	R	R
620 *	E	D	C	1	*C. freundii*	**+**	**-**	R	R	R	R	S	S	S	S	S	S
622 *	F	D	C	2	*E. coli*	**+**	**+**	R	R	R	R	S	S	S	R	R	R
625	A	D	E	1	*E. coli*	**+**	**-**	R	R	R	R	S	S	S	S	S	S
627	F	D	E	1	*E. coli*	**+**	**+**	R	R	R	R	S	S	S	S	S	R
628	F	D	E	2	*E. coli*	**+**	**+**	R	R	R	R	S	S	S	R	S	R
629	E	D	E	1	*K. pneumoniae*	**+**	**+**	R	R	R	R	S	S	R	S	R	R
630	E	D	E	1	*E. coli*	**+**	**-**	R	R	R	R	S	S	S	S	S	S
631	E	D	E	2	*E. coli*	**+**	**-**	R	R	R	R	S	S	S	S	S	S
632	E	D	E	2	*K. pneumoniae*	**+**	**+**	R	R	R	R	S	S	S	S	S	R
633	A	D	E	2	*E. coli*	**+**	**-**	R	R	R	R	S	S	S	S	S	S
634	A	N	E	2	*K. pneumoniae*	**+**	**+**	R	R	R	R	S	S	S	S	R	S
635	B	D	E	1	*E. coli*	**+**	**+**	R	R	R	R	S	S	S	R	S	S
636	B	N	E	2	*E. coli*	**+**	**+**	R	R	R	R	S	S	S	S	S	R
637	B	N	E	2	*E. coli*	**+**	**+**	R	R	R	R	S	S	S	R	S	R
638	C	D	E	1	*E. coli*	**+**	**-**	R	R	R	R	S	S	S	S	S	S
639	C	D	E	2	*E. coli*	**+**	**-**	R	R	R	R	S	S	S	S	S	S
640	D	D	E	1	*E. coli*	**+**	**+**	R	R	R	R	S	S	S	S	S	R
641	D	D	E	1	*K. pneumoniae*	**+**	**+**	R	R	R	R	S	S	S	R	R	S
642	D	N	E	2	*E. coli*	**+**	**+**	R	R	R	R	S	S	S	S	S	R
643	E	D	E	1	*E. coli*	**+**	**+**	R	R	R	R	S	S	S	R	S	R
648	D	N	E	1	*K. pneumoniae*	**+**	**+**	R	R	R	R	S	S	R	R	S	S
649	E	N	E	1	*E. coli*	**+**	**+**	R	R	R	R	S	S	S	S	S	R
650	E	N	E	2	*E. coli*	**+**	**+**	R	R	R	R	S	S	S	R	S	R
651	A	N	E	2	*K. pneumoniae*	**+**	**+**	R	R	R	R	S	S	S	R	R	R

## Data Availability

The data for this study have been deposited in the European Nucleotide Archive (ENA) at EMBL-EBI under accession number PRJEB44418 (https://www.ebi.ac.uk/ena/browser/view/PRJEB44418) (accessed on 1 February 2021).

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
