# Peer review of "Antibiotic-Resistant *Enterobacteriaceae* in Wastewater of Abattoirs"

_antibiotics, 2021, doi:10.3390/antibiotics10050568_

Round 1
Reviewer 1 Report
The manuscript reports important findings about antibiotic-resistant Enterobacteriaceae in wastewater of abattoirs. It requires some revision before its acceptance for publication in Antibiotics.
Abstract
Line 12, As pathogens,
Line 13, as commensals,
Line 16, ESBL, please give full name at its first usage
Lines 18-20, its seems that the sentence is fragmentary.
Line 21, please only give full percentage values (no digits)
Introduction
Line 37, ESBLs are not enzymes, but they excrete enzymes which …
Results
Line 126 and several more times in the text, how is “intermediate” defined?
Discussion
Line 205, the patterns in Figure 1 (469 versus 471; 552 versus 553) look similar, but not identical
Materials and Methods
Line 360, 10 000-fold dilution?
Line 369, AST should be named in full at its first usage
Conclusions
Lines 433-434, … two respects: first, … and, thus, …
References
Please adhere to the formatting rules of the journal, e.g. year in bold, only volumes but no issues to indicate.
Reviewer 2 Report
The study from Homeier-Bachmann is an interesting investigation into Enterobacteriaceae resistance from slaughterhouse samples. I have no major comments, and only minor corrections to suggest:
- ESBL definition (lines 37-39) is narrow
- Please include the catalog number for ‘sterile gauze’ as this is not common practice
- Please expand on the WGS methods, such as trimming parameters other metrics used for QC, contamination, and consensus
- A comparison of sequenced isolates to those previously published would be a nice addition
- The sentence in lines 201-204 is confusing - please shorten / reword as it is not clear what is being said.
- Figure 2 is poor quality and in parts illegible. A heat map would be a better representation of much of this data. At the minimum, text should not be blurry
